

# Modelling Hydrological Ecosystem Services – A state of the art model comparison

Anna Lüke[1], Jochen Hack[1]

[1]Section of Engineering Hydrology and Water Management, Technische Universität Darmstadt, Darmstadt, 64287, Germany

*Correspondence to*: Jochen Hack (j.hack@ihwb.tu-darmstadt.de)

**Abstract.** Different simulation models are used in science and practice in order to incorporate hydrological ecosystem services in decision-making processes. This contribution compares three simulation models, the Soil and Water Assessment Tool, a traditional hydrological model, and two ecosystem services models, the Integrated Valuation of Ecosystem Services and Trade-offs model and the Resource Investment Optimization System model. The three models are compared on a theoretical and

conceptual basis as well in a comparative case study application. The application of the models to a study area in Nicaragua reveals that a practical benefit to apply these models for different questions in decision-making generally exists. However, modelling of hydrological ecosystem services is associated with a high application effort and requires input data that may not always be available. The degree of detail in temporal and spatial variability in ecosystem service provision is higher when using the Soil and Water Assessment Tool compared to the two ecosystem service models. In contrast, the ecosystem service

models have lower requirements on input data and process knowledge. A relationship between service provision and beneficiaries is readily produced and can be visualized as a model output. The latter is especially useful for a practical decision-making context.

## 1 Introduction

The number of tools for assessment, quantification and valuation of ecosystem services (ES) is increasing rapidly. Especially

for water management, modeling of Hydrological Ecosystem Services (HES) is becoming increasingly important (Hack, 2015). Hydrological Ecosystem Services are defined here as proposed by Brauman et al. (2007). The goal of modeling is often the assessment and quantification of ES as well as the mapping of potential supply areas and users of ES. However, hydrological models have a long history in water management and several hydrological models have been used for the quantification of HES (Duku et al., 2015b; Francesconi et al., 2016; Schmalz et al., 2016). In addition to the use of traditional

hydrological models, special simulation models have also been developed in recent years. These models are based on quite different conceptual approaches, but used to answer the same questions and to derive recommendations for policy and planning practice. While several studies using traditional hydrological models (e.g. Duku et al., 2015b; Francesconi et al., 2016; Swallow et al., 2009) or a specific ES model alone have been published (e.g. Hamel et al., 2015; Nelson et al., 2009; Terrado et al., 2014), comparisons of modeling results from both modeling domains (hydrology and ecosystem services) have not so far.



Bagstad et al. (2013), for instance, identified 17 decision-support tools for ES quantification and valuation - all rather recently developed models specifically for the assessment of ES. Bagstad, Semmens, and Winthrop (2013) compare specific ecosystem service models (InVEST and ARIES).

This paper discusses the applicability of three free source models: the Soil and Water Assessment Tool (SWAT), the Integrated

Valuation of Ecosystem Services and Tradeoffs model (InVEST) and the Resource Investment Optimization System model (RIOS) for hydrological ecosystem service modeling in an urbanizing watershed in Nicaragua as a case study example. Thereby not only the biophysical provision of hydrological ecosystem services is modeled, but also potential beneficiaries identified and quantified. The three models are compared regarding their theoretical conceptualization and their practical applicatin potential of the models for decision-making in a policy context (e.g. for the setup of compensation mechanisms for

ecosystem services). The theoretical comparison focuses on modeling approaches and underlying concepts, the assessable HES, the required input data, and meeting of requirements for the modeling of HES. Further on, the three models are applied to the study area in Nicaragua to compare the practical application effort concerning required input data and preprocessing, results, time requirement and training effort, amongst others. The results of the theoretical and practical comparison are evaluated and discussed afterwards. The paper closes with a conclusion presenting recommendations for model application

and future research needs.

## 2 Material and methods

The simulation models chosen for comparison are available without costs, well documented and their use for hydrological ecosystem service modeling has been proven. The Chiquito River watershed in Nicaragua is used as a case study. The concept of hydrological ecosystem services and the policy instrument of payments for hydrological ecosystem services has been applied

in Nicaragua in different contexts and is promoted by the Nicaraguan government to improve the management of natural resources (Hack, 2015; Wheelock Díaz and Jackman, 2007). All modeling data have been retrieved from publically available global databases and socio-economic information provided by Nicaraguan ministries.

### 2.1 Study area and data base

The Chiquito watershed is situated in the Northwest of Nicaragua in the Department León (see Figure 1, top). The river drains over a length of 30 km in East-West direction from the Nicaraguan volcanic chain to the Pacific Ocean. Its source area is located about 2 km Northeast from the city of León, the capital of the equal named municipality. The total area of the watershed is 180.64 km². The watershed is relatively flat, only in the northeastern part elevation increase up to 220 m a.s.l. (see Figure 2, bottom). The Chiquito River traverses the city of León, where the majority of the population of the department lives. An

important tributary to the Chiquito River, the Pochote River, originates at the northern city boundary of León and flows into the Chiquito River shortly after leaving the city. Out of town, the watershed is of rural character with a high use of agriculture.



The watershed is located in the municipality León with a total population density of 2.33 inhabitants per hectare. Since 80 percent of the population lives in the city León, the urban density is 73.41 inhabitants per hectare, whereas the rural population density is 0.48 inhabitants per hectare (Alcaldía Municipal de León, 2008).

The per annual capita water consumption of Nicaragua averages 252 m³. Thereby, households have a per capita consumption

of 37 m³ per year, industry 6 m³ per year and agriculture consumes 209 m³ per capita and year (Food and Agriculture Organization, 2008). The study area has a tropical savanna climate with a pronounced dry season from November to April and a rainy season from May to October, having an average monthly precipitation from 300 to 500 mm. The average temperature varies from 27 to 29° Celsius with the lowest values from December to February (Alcaldía Municipal de León, 2008).

The watershed is characterized by agricultural use, whereby mainly a mosaic of cropland and vegetation dominates, being

grassland, shrubland or forest, interrupted by herbaceous vegetation or forest. In the southwestern part, before the Chiquito River discharges into the Pacific Ocean, the river merges with a permanently flooded forest or shrubland – a mangrove forest known as the Natural Reserve Juan Venado Island of high ecological value and importance for the tourism sector of Nicaragua. The northeastern area of the watershed contains an evergreen forest.

To analyze the watershed's Hydrological Ecosystem Services, the actual situation of the watershed is investigated. Especially,

the main threats in the department León are degradation and erosion of soil. According to Galo Romero (2015), the excessive agriculture and livestock farming cause soil degradation. In the course of this degradation, dust storms arise, which pose a health risk for the population. Therefore, research is focusing on alternative land management practices to maintain soils. Amongst others, these practices can be agroforestry or silvopasture combining forestry with pasture for livestock (Galo Romero, 2015). Water scarcity as well as floods are further risks. Although Nicaragua has large freshwater resources, water

scarcity frequently becomes a problem. For instance, in the western part of Nicaragua, the population was threatened with water scarcity in 2015. Due to high temperatures caused by the phenomena of El Niño, water resources decreased strongly. The water scarcity is mainly related to global climate change resulting in reduced precipitation, but also to the degradation of soils due to inappropriate agricultural use (El Espectador, 2015; Silva, 2015). However, the department León is also vulnerable to floods, especially in the rainy season (López Hernández, 2011).

For model application, publically available data was used that had been retrieved from the internet. Table 1 gives an overview of the required input data for both models, its source, and spatial resolution. Partly, the models require the same basic input data, which have to be preprocessed distinctly and saved in different formats.

The digital elevation model (DEM) used was retrieved from USGS HydroSHEDS and is a void-filled raster, whose no-data voids were filled using interpolation algorithms (Lehner et al., 2006). Before applying the DEM in the models, it was edited

using the ESRI software ArcGIS. Sinks or depressions within the DEM were filled with a hydrology tool of the spatial analyst in ArcGIS to ensure that the flow direction can be derived correctly from the DEM. Furthermore, the original geographic coordinate system of the DEM was reprojected to the UTM coordinate system WGS 84 for the northern hemisphere (Zone 16 N), because the models require raster data in projected coordinate systems. This coordinate system projection is used for all other input data. As can be seen from Table 1, the input raster files had originally different spatial resolutions. To receive



homogenous modeling results, the input raster files are resampled to the DEM's resolution. Therefore, all raster files have a pixel size of 90 m x 90 m.

The GlobCover map contains an error in the land use of the watershed. The city León, situated at the bank of the Chiquito River, is located in the land use raster west of its actual position. To correct the position of León, the tool ARIS Grid & Raster

Editor (ARIS B.V., 2016) is used to change raster values. The location with incorrect raster values is examined in Google Earth to determine the actual land use and land cover. Two hills with bare soil, sparse vegetation, and shrubs were identified in this area. Based on this image, the affected raster cells are changed with the ARIS tool.

The actual position of León is added to the raster by means of a polygon of the extent of the city (Hummel, 2016). This is converted in a raster dataset and mosaiced with an ArcGIS tool to the land use raster. Moreover, the GlobCover map does not

contain roads as land use. Sealed, major roads in the watershed are also added to the raster dataset. For this, a shape file with the main roads of Nicaragua is converted to a raster and mosaiced to the existing land use map.

The input data of the study area used for the SWAT model is shown in

Appendix 1, the input data for the InVEST and RIOS models are shown in

Appendix 2.

## 2.2 Software used to model Hydrological Ecosystem Services

Three different models are applied to the study area in Nicaragua, two special ecosystem service models and one traditional hydrological model. The traditional hydrological model, the Soil and Water Assessment Tool (SWAT) (Arnold et al., 1998; Arnold et al., 2005), was originally developed to predict the impacts of land management practices on water, sediment and agricultural chemical yields. However, several of the model outputs can be used to estimate and quantify the benefits of

Hydrological Ecosystem Services. The models Integrated Valuation of Ecosystem Services and Tradeoffs (InVEST) (Goldman and Tallis, 2009; Sharp et al., 2016) and the Resource Investment Optimization System (RIOS) (Vogl et al., 2016) are developed especially for the modeling of ES by the Natural Capital Project (NatCap). In the following, the three models are briefly described and their specific characteristics highlighted.

The traditional hydrological model Soil and Water Assessment Tool – SWAT was developed by the USDA – Agricultural

Research Service and the Texas A&M AgriLife Research (Arnold et al., 1998; Arnold and Fohrer, 2005). For the present study, ArcSWAT version 2012.10.18 is used as a plugin for ESRI ArcGIS to set up the model and for the preparation of model input data. SWAT is a physically based, semi-distributed, continuous time model to simulate the impacts of land management practices on water, agricultural and chemical yields in large complex watersheds with varying soils, land cover and management conditions over long periods. The main components of SWAT are hydrology, weather, soil temperature and

properties, plant growth, nutrients, pesticides, bacteria and pathogens as well as land management practices (Gassman et al., 2007; Neitsch et al., 2009). SWAT models different physical processes in a watershed using various biophysical input data, such as weather, soil and land use information. To simulate the physical processes, SWAT divides the watershed into subwatersheds, which are further subdivided into hydrological response units (HRUs). These HRUs are lumped areas with





homogenous land use, management conditions, and soil characteristics (Gassman et al., 2007, 1212; Neitsch et al., 2009). The simulation of SWAT is divided into the land and water or routing phase of the hydrological cycle. Whereas the land phase controls the amount of water, sediment, nutrient and pesticide reaching the main channel in each subwatersheds, the water or routing phase directs the movement of the water, sediment etc. through the channel network of the watershed to the outlet

(Neitsch et al., 2009). The simulation of the land phase of the hydrological cycle is based on the water balance equation. Several of the SWAT model outputs can be used to estimate and quantify HES, such as water yield, sedimentation or water quality (Vigerstol and Aukema, 2011). The translation of SWAT model outputs into HES requires post-processing, which is described in the following methodology section. SWAT has mostly been applied to evaluate provisioning and regulating services (Francesconi et al., 2016; Schmalz et al., 2016), such as the freshwater production, water purification and sediment

regulation. Specifically, SWAT can be used to determine a watershed's capacity to provide HES. A main limitation of the SWAT model in HES modeling is that socio-economic data cannot be included and provision and benefits of HES cannot be linked. Therefore, a combination with a socio-economic analysis to compare the modeled service capacity with the societal demand and supply is reasonable (Francesconi et al., 2016).

The ecosystem service model Resource Investment Optimization System – RIOS (Vogl et al., 2016) is an open-source, stand-

15 alone software tool developed by NatCap. For this study, the RIOS version 1.1.16 is used. The RIOS model aims at the determination of locations for management activities to protect, maintain or restore ES, especially HES, in order to generate the greatest benefits for both people and nature focusing on low costs. RIOS is based on a science-based approach operating independently of scale or location and, therefore, it can be used at continental, country, or regional scale. The tool works on annual or longer-term time scales and focuses on land management-based transitions. It uses widely available data on land use

and management, climate, soil, topography, and service demands. RIOS consists of two modules, which are, firstly, the Investment Portfolio Adviser, and secondly, the Portfolio Translator. The Investment Portfolio Adviser determines a most efficient and effective set of investments in management activities with a specific budget, demonstrating where and in what activities investments are appropriate, which is the so-called Investment Portfolio. For this, the Investment Portfolio Adviser uses biophysical and social data, budget information, and implementation costs for different activities. The first step in

generating an Investment Portfolio is the definition of objectives, which the user wants to achieve. RIOS allows the user to select single or multiple objectives with or without weighting. The objectives provided by RIOS are erosion control for drinking water quality or reservoir maintenance, nutrient retention, flood mitigation, groundwater recharge enhancement, dry season base flow, and biodiversity. For the achievement of these objectives, changes in the land management of the watershed may be required. Initial transitions in the vegetation or in land management practices can be caused by different activities. Figure

2 demonstrates that different activities can cause the same transitions, but they may have varying costs and be applied in distinct areas of the watershed. These transitions influence directly or indirectly hydrological processes and biodiversity. The transitions included in RIOS are: keep native vegetation, revegetation (assisted or unassisted), agricultural vegetation management, ditching, fertilizer and pasture management. Whereas these transitions are defined in the software, the selection





of activities is determined by the user. This means that RIOS does not assist in the selection of activities, but identifies where the selected activities obtain the greatest returns towards the user's objectives (Vogl et al., 2016).

Then, RIOS uses a ranking model to determine the areas, where investments have the highest return on investment. The approach bases on the condition that a limited set of biophysical and ecological factors determine the effectiveness of each transition in achieving each selected objective. Furthermore, a subset of landscape factors is defined having an impact on the effectiveness of activities and reflecting the landscape conditions, and finally affecting each objective. The model approach assumes that the conditions of the surrounding landscape mainly determine the impacts of the transitions. Thereof, RIOS determines ranking scores for each user-defined spatial unit, the so-called pixel, derived from cell sizes of the input grid raster. Four components, being the conditions of the pixel itself and the conditions of the surrounding area are the determining factors for these scores (Vogl et al., 2016).

The Portfolio Translator creates three major scenarios displayed as land cover maps and basing on the Investment Portfolio. The first scenario (baseline) contains the current land cover. The second scenario (transitioned) demonstrates new land cover combinations and protected areas caused by the implemented activities. The third scenario includes implemented activities, but former protected areas are degraded demonstrating their benefits, when they are protected (Vogl et al., 2016).

The ecosystem service model Integrated Valuation of Ecosystem Services and Tradeoffs – InVEST (Sharp et al., 2016; Tallis and Polasky, 2009) is a set of different models to quantify and map ES. It is, as well as RIOS, an open-source, stand-alone software developed by NatCap. InVEST aims at the assessment of land cover changes on different ES in large watersheds comparing alternative land use scenarios. It aims to inform decision makers in natural resource management and to point out the impacts of changes in ecosystems to the benefits of people. The general approach of the spatial explicit models of InVEST bases on production functions to quantify the impact of changes in the functions and structures of an ecosystem on the flows and outputs of ES. These functions are simplifications of common hydrological relationships (Vigerstol and Aukema, 2011). InVEST calculates the results annually, based on land use information. The inputs are spatially explicit georeferenced raster or shape files and tables containing coefficients for each land cover type. The model calculates on pixel basis, breaking up the watershed into pixels pursuant to the spatial resolution of the input data. These pixel results are aggregated to subwatershed results in further modeling steps. Since the spatial resolution is flexible, InVEST is capable to model at local, regional or global scale (Stanford University et al., 2016; Vigerstol and Aukema, 2011). To visualize the outputs of intermediate modeling steps, final service levels and economic estimates, a mapping software or geographic information system (GIS) is necessary (Stanford University et al., 2016). The set of InVEST models can be used to quantify and map terrestrial, freshwater, and marine ecosystems. They can be categorized into three groups, which are supporting services, final services, and tools to facilitate ES analyses (Sharp et al., 2016). The category of supporting ecosystem services contains, for example marine water quality and habitat quality. Final ecosystem services can be modeled directly or using the biophysical supply. Coastal exposure and vulnerability as well as nature-based recreation and tourism are services modeled directly. In contrast, services like water purification and climate regulation are modeled by means of their supply. Exemplarily, the supply of nutrient and sediment retention is used to determine the final service of water purification. Carbon storage and sequestration can be used to conclude



on the service of climate regulation (Sharp et al., 2016; Stanford University et al., 2016). The models mentioned above are merely a selection of examples. Further information on the available set of InVEST models can be extracted from Sharp et al. (2016) and Stanford University et al. (2016). For the presented study, two models of InVEST are selected: the Water Yield Model and the Sediment Delivery Ratio (SDR) Model.

The Water Yield Model estimates the annual average quantity of water provided by a watershed. This can be used, for instance, to evaluate potential hydropower production in a watershed. The results of the model illustrate, which areas have the highest contribution to water yield and, thus, to hydropower production. Therefore, the Water Yield Model can assess different land use scenarios and their impacts on water yield. The model calculates the relative contribution of each land parcel to the annual average hydropower production, instead of directly modeling the effects of land use changes on hydropower (Sharp et al.,

2016; Stanford University et al., 2016). This calculation, based on a gridded map, is divided into three steps, as illustrated in Figure 3. Firstly, the amount of water flowing off each pixel, which is the amount of precipitation less evapotranspiration, is calculated. Surface, lateral and base flow are not considered differentially. The pixel runoff is then summed up and averaged to subwatershed level. This is because the theory of the Water Yield Model is developed at subwatershed to watershed scale. Therefore, the results are only reliably interpretable on subwatershed to watershed scale. In the second step, the amount of

surface water, which is used for hydropower production, is determined by subtracting water, consumed for other purposes, by the water scarcity model. The results of this step may be used to assess the possible water supply of the subwatershed and to determine whether water is scarce. Thirdly, the energy produced by the water reaching the reservoir and the energy's value may be estimated. In general, the Water Yield Model bases on a simplification of the water cycle mainly including precipitation, transpiration, and evaporation (Sharp et al., 2016).

The underlying model equations can be extracted from Sharp et al. (2016). The input data required for the Water Yield Model consists of different GIS raster datasets with values for each cell, shape files containing watershed and subwatershed polygons, and tables in CSV-format. The tables contain biophysical coefficients for each land use class, a demand table comprising consumptive water use of each land use class and hydropower stations containing specific information. The output is divided in intermediate results per pixel and final outputs at subwatershed level. The final outputs are in shape file format containing

a table with the calculated values per subwatershed, such as the volume of water yield, the total water consumption, the total realized water supply volume for each subwatershed. If the hydropower valuation model is used, the table contains additionally the amount of energy produced and the value of the landscape per subwatershed to provide water for hydropower production over a specified time (Sharp et al., 2016).

The Sediment Delivery Ratio Model, in short SDR Model, estimates the overland sediment generation and its delivery to the

stream. The results of the model illustrate the ES of sediment retention in a watershed, which is significant for water quality and reservoir management. Changes in land use or alterations in land management practices can influence sediment export in a watershed. The SDR Model focuses on overland erosion processes. The biophysical model is spatially explicit and adopts the spatial resolution of the input DEM raster. Firstly, the model calculates the annual loss of each pixel with the Revised Universal Soil Loss Equation (RUSLE). The model determines the sediment delivery ratio (SDR) in two steps basing on a





function of upslope area and downslope flow path, which is illustrated in Figure 4. In the first step, a connectivity index is calculated, from which, in a second step, the sediment delivery ratio is derived for each pixel. With the sediment delivery ratio and the amount of annual soil loss calculated with the RUSLE equation, the sediment load is determined. The SDR Model requires, such as the water yield model, different biophysical input datasets in georeferenced raster, shape file and table format.

The required raster datasets are a DEM, the rainfall erosivity index, the soil erodibility, and the land use and land cover. Optionally, a drainage layer can be used to include artificial connectivity to a stream, such as urban areas or roads. The outputs of the SDR Model are divided into intermediate and final results. The intermediate results are raster sets containing results per pixel, which should not be used for an evaluative interpretation because the model assumptions are based on processes at subwatershed level, and results per watershed or subwatershed in a shape file. This shape file contains the total amount of

sediment exported to the stream, the total amount of potential soil loss, and the sediment retention in tons per watershed. The outputs of the SDR model, being the annual sediment load delivered to the stream and the amount of sediment eroded in the watershed and kept by vegetation and topography, can be used to evaluate the ES of sediment retention. For a quantitative valuation, the model computes the sediment retention as the difference to a hypothetical watershed of bare soil. Moreover, an index of sediment retention is calculated, identifying areas that contribute more to retention with reference to bare soil. This

index can be used for a qualitative assessment (Sharp et al., 2016).

### 2.3 Methodology - Application of the models, data pre- and post-processing

These three models are compared theoretically and practically (in application to a case study) in the following to evaluate their suitability for decision-making. The theoretical comparison bases on different qualitative and quantitative criteria to highlight the differences or similarities of the models. These criteria are the model type, the original model purpose and the general

model concept, as well as the model approach reflecting the structure of the model. Furthermore, the underlying equations are considered as well as the temporal and spatial resolution and the scale of the results. Besides, it is examined, which ES are assessed and whether it is possible to include beneficiaries. Another point of the theoretical comparison considers the mapping and the displaying of multiple ES. Moreover, the model limitations are compared as well as the required input data. The theoretical comparison is complemented with the practical application of the three models for a study area in Nicaragua.

Furthermore, the results of the practical application are compared concerning different criteria. This comparison focuses on the application and the results of the models as well as on the model application effort. The qualitative and quantitative criteria of the practical comparison are the application objective, which can be achieved with the model, the types of results, the kind of visualization, how and whether beneficiaries are included and areas of provision and use of ES can be distinguished. Another criteria is the possibility to combine the models with each other. Since uncertainty is an important point in modeling, this issue

is also considered in the comparison to show to what extent and how the models deal with this issue. Moreover, the effort to apply the three models to the study area is compared. The evaluation criteria here are data requirement and necessary preprocessing, data availability, training effort to apply the model and the presence of instructions or user manuals. Furthermore, the time, required to apply the three models, is compared.



In addition to the theoretical comparison of the models and their underlying structures, they are applied to a study area in Nicaragua. Since the three compared models vary in their theoretical concepts and structures as well as in their application, different approaches are necessary for the practical comparison. As can be seen from Figure 5, the general approach for the models InVEST and RIOS is the same. The output of InVEST is in shape file format and can be displayed by means of a GIS

representing the capacity, demand, and supply of a specific ES. RIOS generates an Investment Portfolio, containing raster files and tables with the results. These are the locations for activities to restore or protect the selected ES, the budget spent and alternative land use scenarios, amongst other information. The results can be visualized with a GIS. Therefore, InVEST and RIOS require no post-processing of their outputs. In contrast, the application of SWAT requires post-processing to translate the model outputs in ES, since the original purpose of SWAT is the prediction of impacts of land management practices on

water, sediment and agricultural chemical yield in large watersheds. The visualization of the results requires for all three models the use of a GIS.

Several of the SWAT model outputs can be used to estimate and quantify the capacity of ES. For the study area in Nicaragua, the HES of water flow regulation and sediment retention are analyzed with the three models. Therefore, appropriate indicators form the SWAT model output have to be selected. These variables can be used to indicate HES either in combination or alone.

There are different approaches to translate SWAT outputs into HES. The approach for the study area in Nicaragua follows Schmalz et al. (2016). Table 2 shows the variables selected to represent the ES of water flow regulation and sediment retention. As can be seen from Table 2, the water flow regulation is composed by variables reflecting the water cycle, which are the surface and lateral flow, the soil water content, and the groundwater contribution indicating the quantity of water retention and the retention capacity of the surface and underlying soils. To represent the HES of sediment retention, the sediment yield

transported to the main channel is selected, which can be used to indicate areas with little sediment export and, thus, a high sediment retention. For mapping of ES, the hydrological response unit basis is suitable due to the finer spatial resolution compared to the subwatershed, what enables a more detailed display of the results.

The output of each variable is averaged over the last five years of the simulation period, which is from 2008 to 2013, to compensate annual variabilities. The outputs of each variable are assigned to a ranking scale from one (very low potential to provide ES) to five (very high potential) to make different HES comparable, proposed by Burkhard et al. (2014). Each variable

is subdivided into value ranges assigned to the ranking scale, using the statistical data mean, maximum, and minimum to create class breaks. To receive the potential of water flow regulation, all ranking values of all variables are averaged. The result is a ranking value for each HRU reflecting its capacity to provide water flow regulation. Since the sediment retention is determined with one SWAT output variable, it is not necessary to average the ranking values. The ranking values are visualized on HRU

basis using a GIS.





## 3 Results

The theoretical comparison focuses on the theoretical model fundamentals and consists of the comparison points described in the methodology section. The following section focuses on the results of the practical application to the study area in Nicaragua.

As described in the methodology section, different variables of the SWAT output are selected and post-processed to display HES. Figure 6 shows the results generated with SWAT, demonstrating the capacity of the watershed to provide the HES of sediment retention and water flow regulation. These functions are only HES, when people are present who potentially could benefit from them. Since it is not possible to integrate social data in the SWAT model, a visual comparison is performed, illustrated in Figure 6 (right side). Therefore, the capacity maps are compared with the beneficiaries' rasters created for the RIOS model. Since 80 percent of the population in the watershed lives in the city of León, this area is strongly weighted. Therefore, a second raster is created containing only the rural population. For the visual comparison of the SWAT results, a raster is created containing the population per HRU. As can be seen from Figure 6 (top left), large areas of the watershed provide a very high capacity to retain sediment. However, sediment retention especially takes place in the northern part of the watershed with a low number of beneficiaries. In the eastern part of the watershed, an area with high retention and a medium number of beneficiaries is located. This is also a benefiting area (see beneficiaries rural) of water flow regulation (bottom left) with a medium to high capacity. Furthermore, the people in the east of León (see beneficiaries all) benefit from a high water flow regulation. In contrast, areas with a very high water flow regulation located in the south-west of León, have a low to medium number of beneficiaries (see beneficiaries rural).

Since RIOS consists of two modules, the Investment Portfolio Adviser and the Portfolio Translator, there are two different outputs. RIOS determines areas for the implementation of activities to restore or maintain different ES or objectives. For the study area in Nicaragua, the objectives erosion control for drinking water quality, flood mitigation and dry season base flow are selected. RIOS determines areas, where activities to achieve these objectives are best situated, with regard to the benefits of people, nature, and costs. Therefore, one result of RIOS is a budget report showing the costs and the converted area for each activity. Because there are two different raster images representing all and rural beneficiaries, two simulations of RIOS are run. However, the results are very similar, only the results for all beneficiaries are presented here. Figure 7 shows the implemented activities. Mainly, grass strips, protection and in some areas, agroforestry is implemented. Grass strips are only allowed on areas with a slope smaller than twelve percent. Therefore, in the steeper areas agroforestry is prioritized. Protection is focused in areas with native vegetation, which is tropical evergreen forest and swamp forest. Reforestation is implemented at shrub areas. Riparian management is mainly chosen for the western part of the watershed in downstream areas.

The second output of RIOS, generated by the Portfolio Translator, contains three land cover scenarios. These are the baseline scenario with the starting land cover, the transitioned scenario showing the transitions in land cover caused by the activities, and an unprotected scenario indicating what changes in land cover are expectable, when protected land use classes are degraded. Figure 8 visualizes the land use classes, which are changed by the implemented activities.



The impacts of the implemented activities by RIOS on the HES can be evaluated with other models, for example with the InVEST Water Yield Model using the generated land use scenarios as inputs. This is shown in Figure 9, which contains the results of InVEST for the three land use scenarios generated by RIOS. As can be seen from Figure 9, the implemented activities have an impact on the water yield of the subwatersheds. The subwatersheds 4, 8, 14, 15, 17, 18, 20, and 21 of the transitioned scenario have a greater water yield than the baseline scenario. Considering the objective of base flow enhancement and drinking water supply, this is beneficial. In contrast, an increase in water yield may also be adverse considering flood mitigation. The third scenario includes the transitioned areas, but formerly protected areas are degraded. The degradation of these areas influence the water yield of the subwatersheds located in this areas. These are the subwatersheds 2, 3, and 23 having an increase in water yield due their reduced water retention capacity. However, the results are annually and do not reflect seasonal variability, for example due to rainy and dry seasons.

Furthermore, the InVEST Water Yield Model is run with the original input data, additionally using a water demand table to analyze water scarcity. Figure 10 shows the results of this run, illustrating the available water yield per subwatershed, the water consumption per subwatershed and the water supply per subwatershed, which is the available water yield minus the water consumption. According to Figure 10, the subwatersheds with the highest provided water yield are 6, 2, 5, and 8. The water consumption is highest in the urban area of León (subwatersheds 6, 9, 10, and 12). However, a comparison of the maximum values of water demand and the values of water yield shows that the maximum of water demand (1 851 418 m³ per subwatershed) lies in the lowest interval of the water supply. Therefore, it can be assumed that the water yield exceeds the demand, what is visualized in the last image of Figure 10. A relevant decrease in the water amount can only be determined in the subwatersheds 4 and 12. However, it should be taken into account that the results are annual averages, again not representing seasonal variability.

Furthermore, the InVEST Sediment Delivery Ratio model or SDR model is applied for the study area in Nicaragua. The SDR model estimates sediment export and retention for each subwatershed, as illustrated in Figure 11. The sediment export in tons per subwatershed (left figure) is highest in the subwatersheds 8, 16, 20, and high in the numbers 6, 14, 15, 21, and 23, what is similar to the result of SWAT (see Figure 6). The sediment retention (right figure), expressed in tons per subwatershed, is estimated in reference to a degraded watershed with bare soil. The value of sediment retention is based upon the difference between sediment export from the bare soil watershed and the input scenario. This may be the reason, that sediment retention is highest in the subwatersheds 16 and 14, 15, 20, and 23, although these are the subwatersheds with the highest sediment export to the stream. Due to the calculation of sediment retention in reference to a bare soil watershed, these subwatersheds are considered as retention areas, because the soil loss by the current land use is far less than the soil loss of a bare soil watershed.

As can be seen from the results, the three models can be used to achieve different objectives. SWAT allows analyzing the capacity of HES in detail. In contrast, InVEST gives a quick overview of different ES. RIOS can be used to determine activity areas for the protection and restoration, especially, for HES.





# 4 Discussion

The comparison of the three models shows that differences between their methodological approaches and results exist. The modeling results reveal that the different models are not directly comparable. The differences of the models, their results, and the specific application effort, in reference to the practical application are summarized in Table 3 and discussed in the following.

While SWAT and InVEST are based on similar conceptual approaches to model hydrological processes, RIOS follows a different approach, determining scores for particular activities using biophysical input data. However, the SWAT model is more complex, and thus it models hydrological processes in more physically-based detail. This requires a large number of input data in comparison to the other models. Since the RIOS model bases on a ranking approach combining biophysical and social input data, it does not model underlying hydrological processes and HES. It uses input data to determine activity preference areas for HES.

The application effort for the SWAT model is particularly higher. This is due to the high time requirement for the preprocessing of input data, the training effort to apply the model and the necessary post-processing to quantify and visualize HES. Since different variables of the SWAT output are selected as indicators of a specific HES, the variables are transformed to a ranking scale (five classes), reflecting relative service capacity. The chosen class limits have a strong influence on the result. The display of spatial variabilities can be affected by the chosen class limits. Other approaches to model HES with SWAT modify and complement the model. This requires specific knowledge and thus, complicates the application for decision makers. The InVEST and RIOS models are more user-friendly due to their fewer input data requirements and the lower training effort to apply the models.

A main difficulty in all model's application is data availability. Especially, for the SWAT model the required data is not widely (publicly and free of charge) available. The reason for this is on the one hand, the complexity of the model requiring a large amount of different variables, on the other hand the location of the study area in Nicaragua. The data situation in developing countries is mostly poorer than in industrialized countries. The data situation is particularly high in the United States, especially for the SWAT model, which is developed and commonly applied there. Furthermore, available data is often not readily usable in the models and needs preprocessing. Additionally, open-source data, which is globally available, has a coarse resolution. This is particularly adverse for the modeling on a regional or local scale (< 100 km²), because land use and land cover data is displayed in a coarse resolution, not reflecting the heterogeneity of land use patterns appropriately. Recommendations for local land use improvements remains therefore challenging. Since InVEST and RIOS are developed for decision makers and with focus on an application in developing countries, they use more commonly available data. However, the application of the RIOS model needs a table, containing user-specified activities, to cause the desired transitions. Hence, previous knowledge of the watershed or the target area is required to determine appropriate activities. It is advantageous, thus, to know, which problems or HES occur in the area to define activities and specify the land use classes suitable for the respective activity. For this, local knowledge or the application of a model, which maps HES, is helpful.



Considering these points, recommendations for the application of the models can be derived. The SWAT model, a traditional hydrological model, is generally suitable to quantify and map the capacity of HES. Since it is not able to include data regarding potential beneficiaries of HES and its focus on hydrological processes, post-processing for the display of HES is required. This complicates its immediate application for decision makers. However, a main advantage of the model is that it bases on widely

accepted and applied hydrological process knowledge and thus, simulates these processes more detailed and, at least partly, physically-based. The model complexity and the large amount of required input data is detrimental for the application in decision-making processes. Therefore, SWAT is recommended for a detailed analysis of specific HES, if sufficient data, time, and hydrological expertise are available.

In contrast to SWAT, RIOS and InVEST can be applied in situations with a limited availability of data and time. Since the

approach of InVEST is simpler than the concept of SWAT, it can be used to give a quick overview of different HES. Furthermore, it allows partially an economic evaluation and is capable to model service demand and supply. The disadvantage of InVEST is that the results, up to now, are only reliable on subwatershed basis, not reflecting finer spatial patterns. Besides, it calculates on annual basis, thus seasonal variability is not considered. This is a clear disadvantage when ES also follow a seasonal variability.

The RIOS model is especially useful to locate activity preference areas for the maintenance and restoration of multiple HES. The medium data and time requirement facilitates its application for decision makers. A difficulty is the determination of appropriate activities and their implementation on different land use classes. Furthermore, the impacts of the implemented activities cannot be evaluated with the RIOS model itself. Therefore, it may be useful to combine RIOS with other models, for example with InVEST or SWAT. It is also possible to use SWAT to show the impacts of different land use scenarios provided

by RIOS. The results of InVEST and SWAT can help to define the activities and choose the objectives in RIOS. Figure 12 illustrates a possible combination of the models. RIOS can be used with InVEST or SWAT in an iterative process, for example to determine activity preference areas for a water fund and to evaluate the scenarios of alternative land use and management scenarios. For this, InVEST is especially suitable, because it requires almost the same input data as RIOS. Moreover, the outputs of RIOS can serve as direct input for InVEST. Furthermore, the results of InVEST or SWAT can assist in the selection

of activities in RIOS. Therefore, a combination of the models may help decision makers to get a good overview of the HES provided in a watershed and where activities to restore or protect them can be implemented.

## 5 Conclusion

The development of the ES concept and associated with this, the incorporation of ES in decision-making has led to a large amount of tools and models to support decision makers. The three models compared here are based on different modeling

concepts. The software SWAT models hydrological processes, based on widely accepted hydrological equations and process knowledge, with high complexity. Since SWAT simulates the underlying processes, appropriate indicators have to be chosen from the model output to quantify HES. InVEST uses ecological production functions, basing on simplified hydrological



processes, to quantify and map several ES. In contrast, RIOS follows a different concept, determining activity areas with a ranking approach. For this, RIOS uses socio-economic data and biophysical input data, which represents the landscape context and thus, the components of hydrological processes. There are several differences in the modeling approaches, but also similarities, e.g. between SWAT and InVEST. These differences and similarities emerged more detailed in the practical

comparison and the application of the three models to a watershed in Nicaragua. The practical application demonstrated that the SWAT model requires a high application effort due to its large amount of input data and the necessary post-processing to visualize HES. However, it delivered detailed results on HRU basis, reflecting the spatial variability of HES capacity throughout the watershed. In contrast, the InVEST model required a medium effort, because it depends on less and better available input data. Since it is able to incorporate partially socio-economic data, it was able to model, in addition to service

capacity, water yield service demand, and water supply at subwatershed level. Therefore, InVEST offers a relatively fast and easy option to model HES. Slightly different results were produced by RIOS, which mainly requires the same biophysical input data as InVEST. In contrast to the other two models, RIOS does model neither hydrological processes nor HES, but determines activity areas, in this investigation to maintain and restore erosion control, dry season base flow, and flood mitigation, with a medium application effort.

Since the inclusion of beneficiaries is essential to determine HES, future research may develop more tools to display service provision and use areas based on more detailed hydrological process knowledge. The InVEST and RIOS models represent already a good approach for the inclusion of beneficiaries. Other approaches, like the probabilistic ARIES model (Villa et al., 2014) simulate and visualize areas of service provision, benefits, and flow paths between the use and source regions. However, ARIES can only apply biophysical relationships if sufficient data is available. Unless it relies on probabilistic relationships

from data of other sites to link spatial input data and ES values.

In summary, provisioning and regulating HES can already be modeled in a useful way. It is also possible to visualize service demand and supply and to include beneficiaries of HES. However, the model application in decision-making processes remains challenging, mainly due to high application effort. Since the quantification and mapping of HES represents a good opportunity for the protection of ES and the maintenance of natural resources for human well-being, future research in HES modeling

should dedicate to the development of reliable, easy applicable tools, which base on hydrological process knowledge, incorporate the beneficiaries of services, and require few and widely available input data. Furthermore, research in HES modeling should address the uncertainty of models or rather estimates of uncertainty with respect to the outputs to avoid a false confidence in results. This is also important, concerning the poor data situation for many regions, which does not permit a calibration of the models. For future HES research, the data situation and availability should be improved. The

implementation of monitoring programs could help to improve the data situation and to improve the knowledge on processes that influence water flow and quality, except of land use and land cover. Therefore, additional research ought to be dedicated to other influence factors on HES than land use and land cover.



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



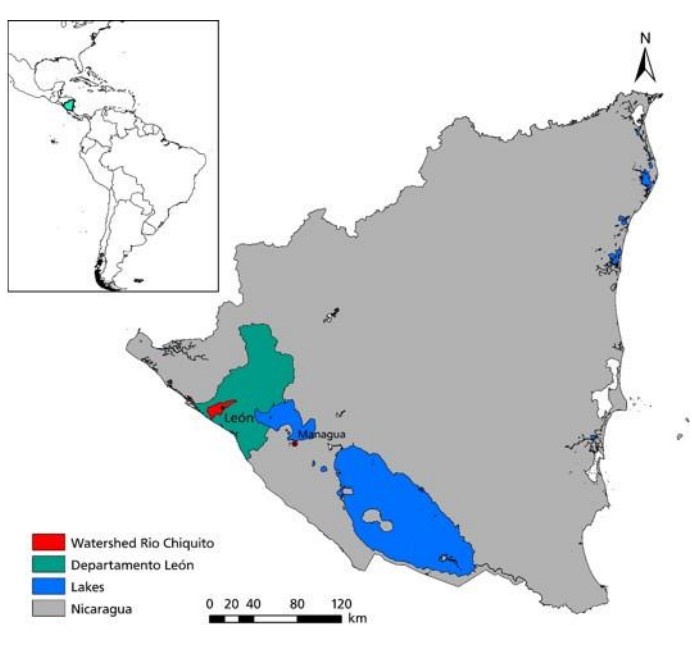

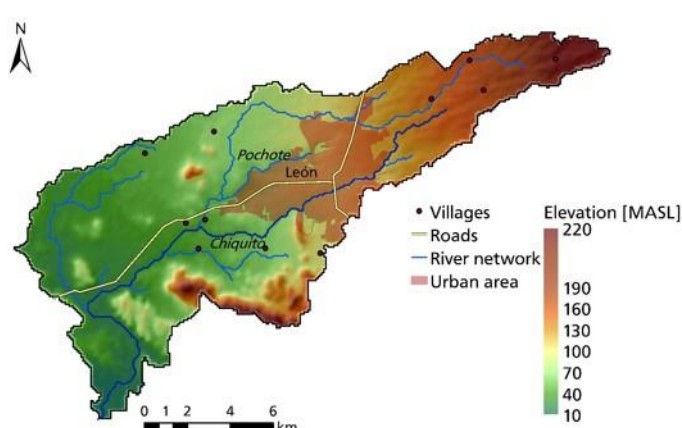

Figure 1: Location of the study area (top) and topography of the Chiquito watershed (bottom).





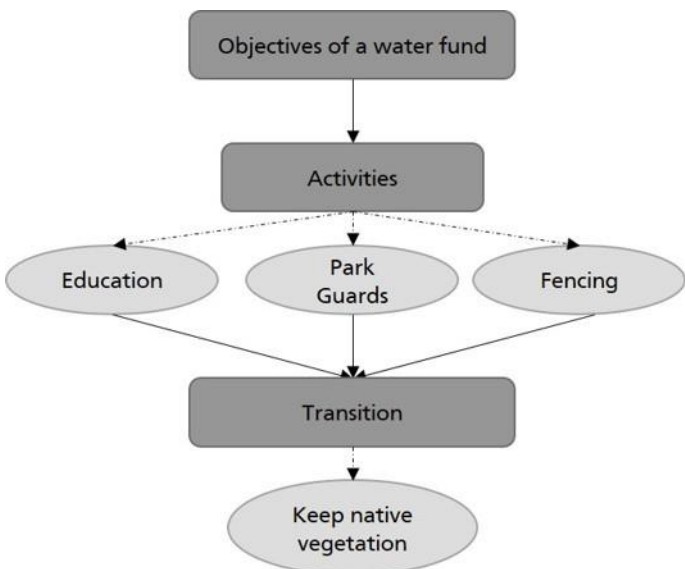

**Figure 2: Relationship between activities and desired transitions in a watershed. Here an example for a water fund. (Vogl et al., 2016).**

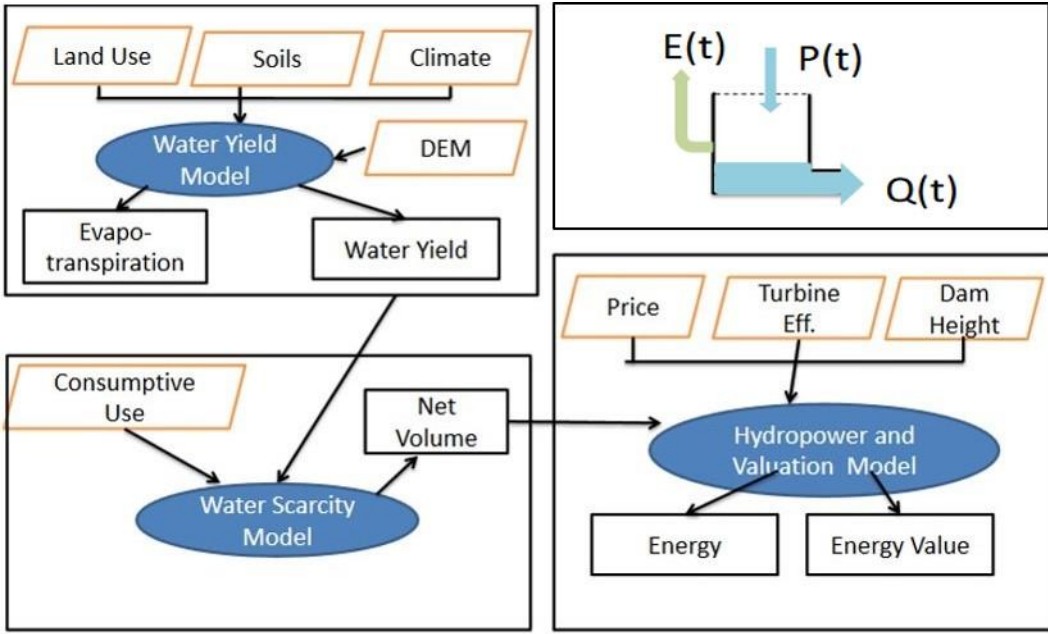

5    **Figure 3: Structure of the Water Yield Model (Guerry et al., n.d.).**





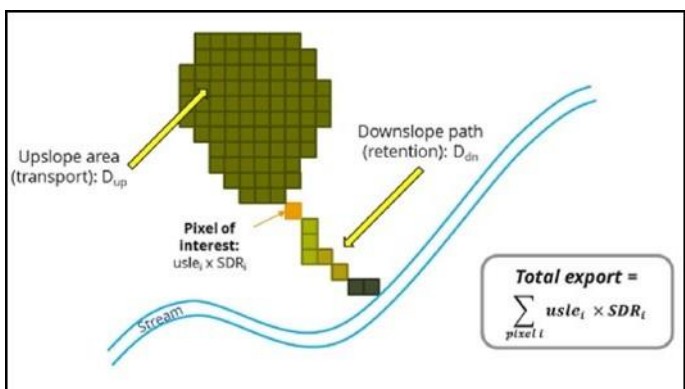

**Figure 4: Sediment delivery ratio of InVEST (Sharp et al., 2016, 149).**

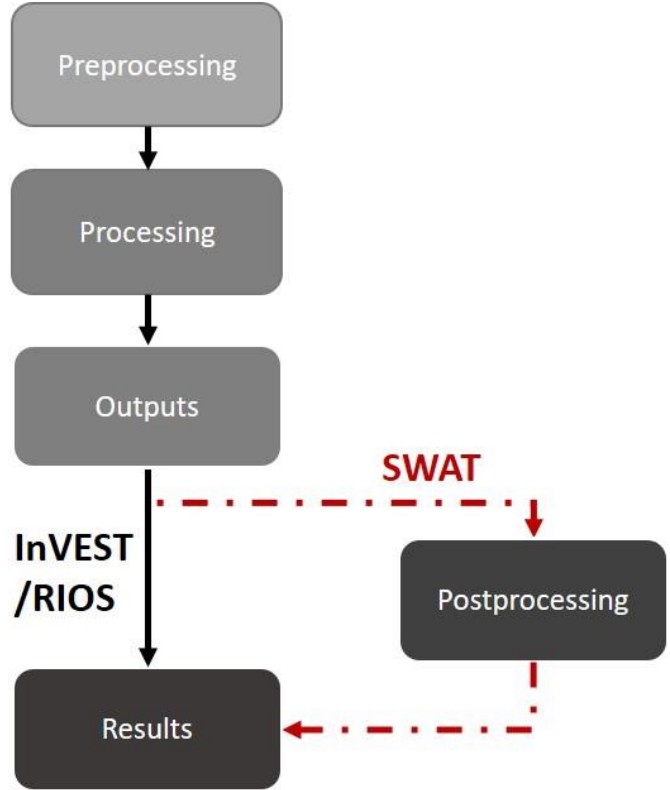

**Figure 5: Application approaches for the three models.**



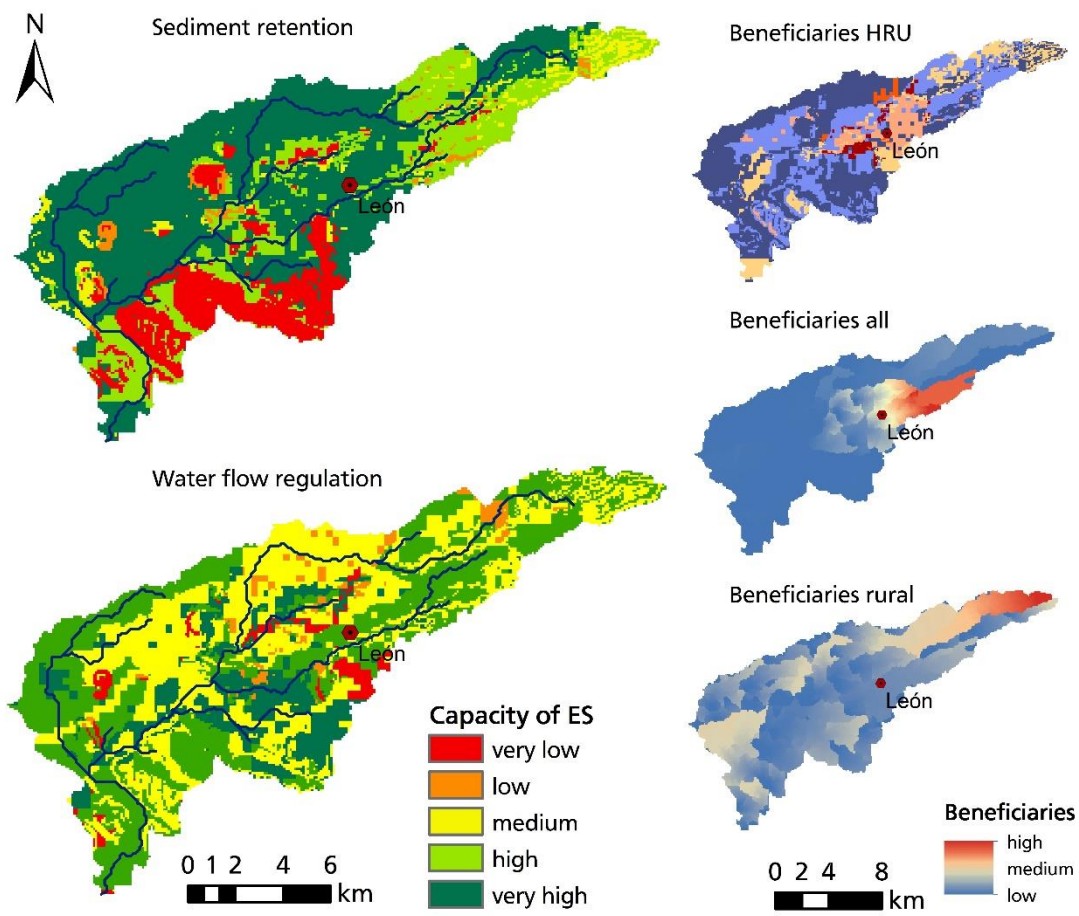

**Figure 6: Modelling results of SWAT.**





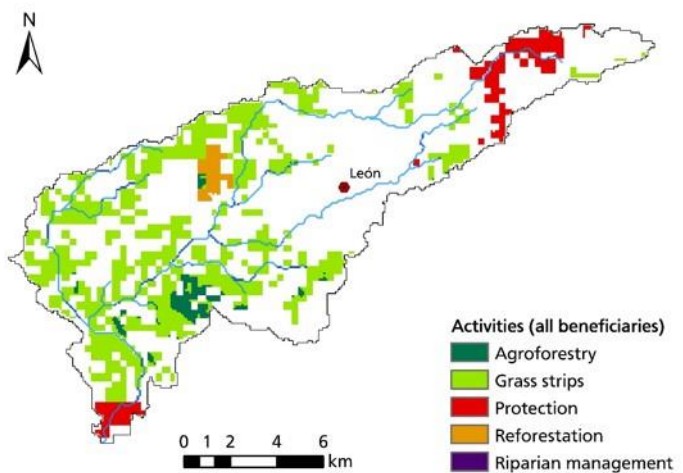

**Figure 7: Modeling results of the RIOS Investment Portfolio Adviser.**

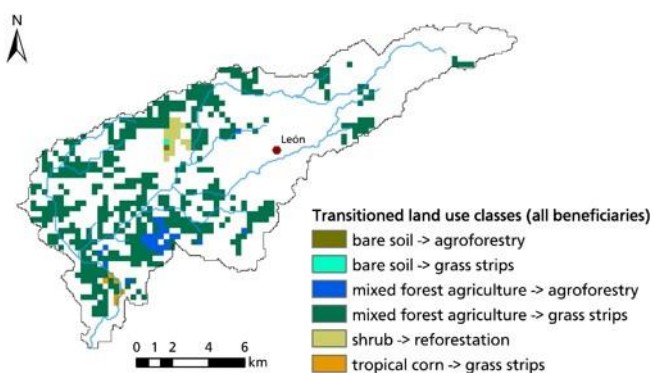

**Figure 8: Modeling result of the RIOS Portfolio Translator.**

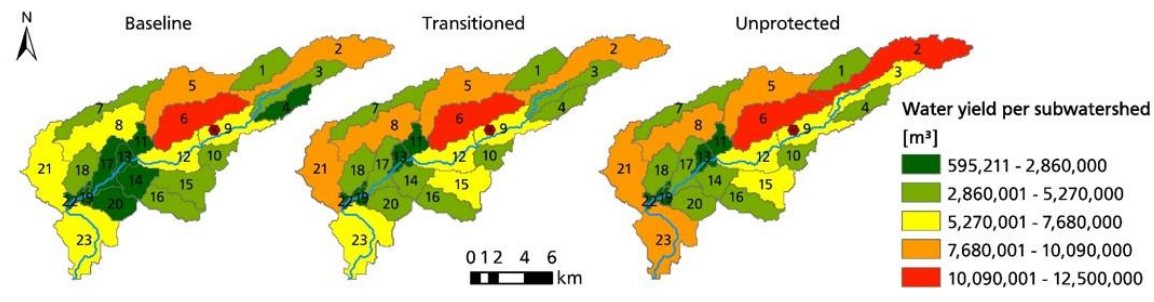

**Figure 9: Modeling results of RIOS evaluated with the InVEST water yield model.**



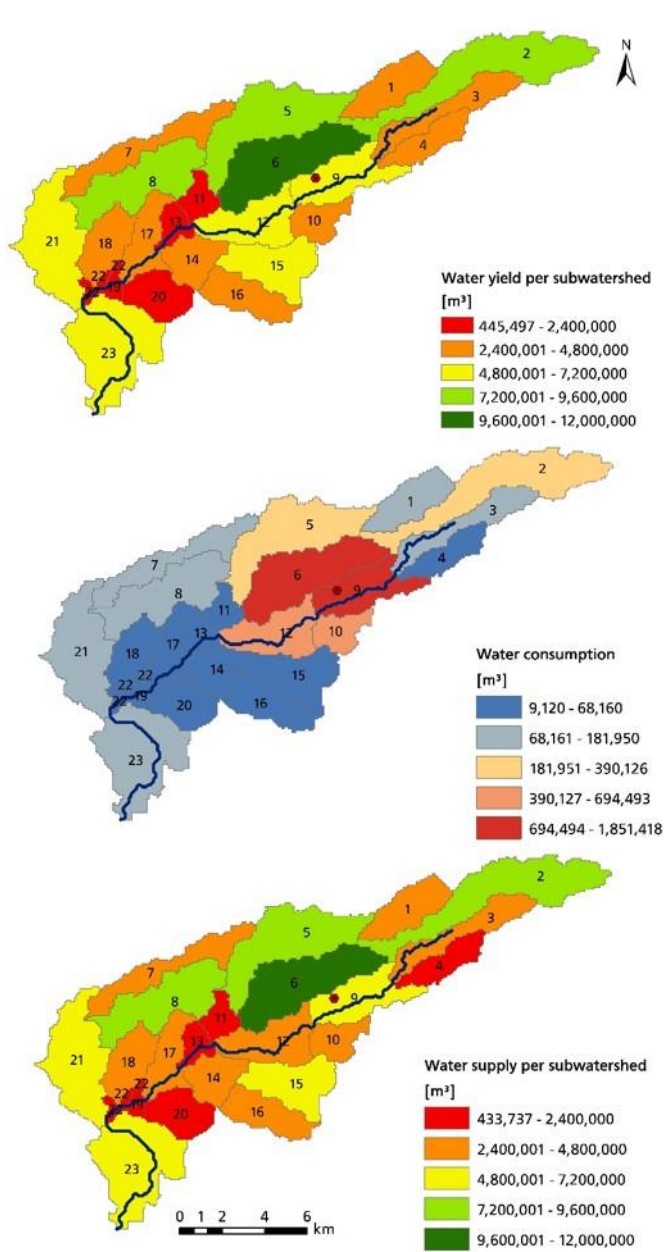

**Figure 10: Output of the InVEST Water Yield Model.**





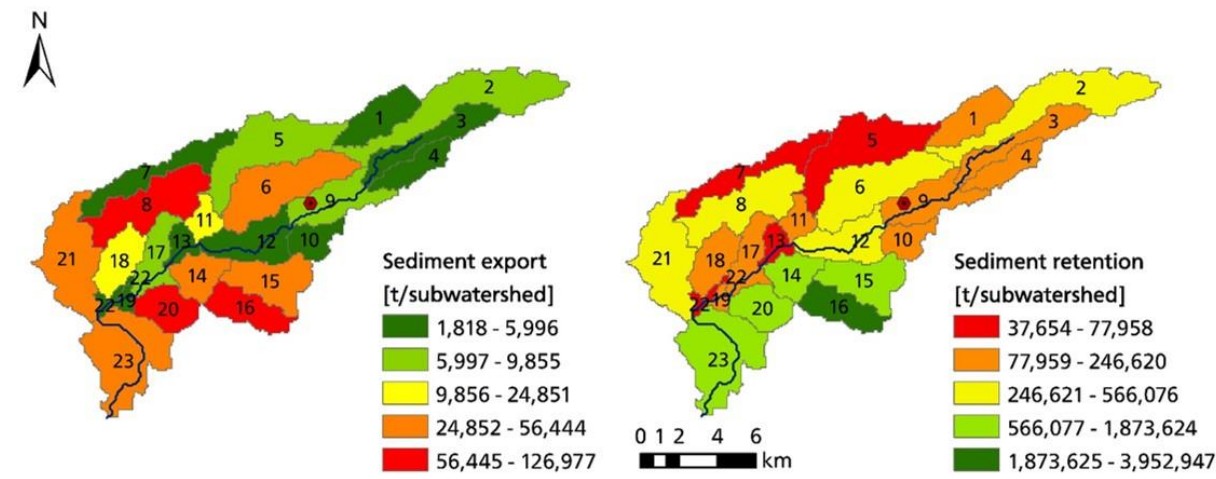

**Figure 11: Output of the InVEST SDR Model.**

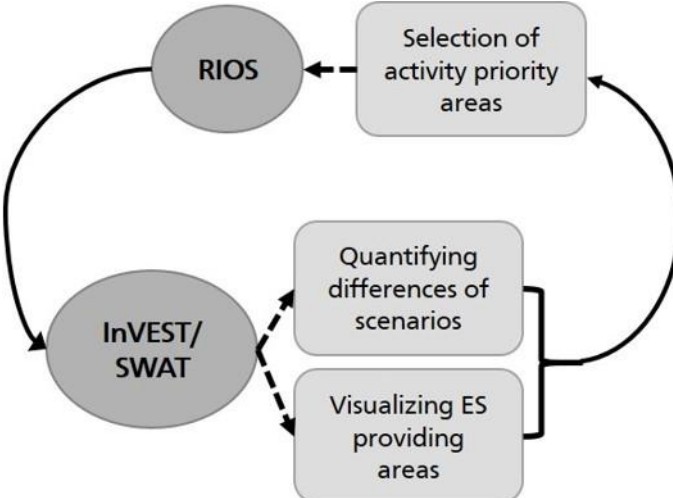

**Figure 12: Possible iterative combination of models.**

5   **Table 1: Model input data specification.**

| Data type (format) | Source | Resolution | Required for: |
|---|---|---|---|
| *Biophysical inputs* | | | |
| Digital elevation model (raster) | USGS HydroSHEDS (Lehner et al., 2006) | 3 arc-seconds → approx. 90 m at equator | InVEST, RIOS, SWAT |
| Land use/land cover (raster) | GlobCover 2009 (ESA and Université | 300 m | InVEST, RIOS, SWAT |





| Data type (format) | Source | Resolution | Required for: |
|---|---|---|---|
| | Catholique de Louvain, 2010) | | |
| Weather data (gages) | Data from NCEP for 1979-2014 (Texas A&M University and Texas A&M AgriLife Research, 2016) | 2 weather stations | SWAT |
| Soil data (raster and database) | HWSD, (Food and Agriculture Organization of the United Nations et al., 2012) | 30 arc-seconds → approx. 1 km | InVEST, RIOS, SWAT |
| Average annual rainfall (raster) | WorldClim, annual average from 1960-1990 (Hijmans et al., 2005) | 30 arc-seconds → approx. 1 km | InVEST, RIOS |
| Mean annual actual evapotranspiration (raster) | CGIAR-CSI, annual average from 1950-2000 (Trabucco and Zomer, 2010) | 30 arc-seconds → approx. 1 km | RIOS |
| *Socio-economic input* | | | |
| Location & number of beneficiaries (raster) | Calculated with population density (Alcaldía Municipal de León, 2008) | Municipality León | RIOS |
| Per capita water consumption (table) | FAO Aquastat data (Food and Agriculture Organization, 2008) | Annual average for Nicaragua | InVEST |

**Table 2: Variables to represent the selected HES (Arnold et al., 2012).**



| Ecosystem service modeled | Variables of the HRU output file | |
| --- | --- | --- |
| | Variable name | Definition |
| Water flow regulation | SW_END | Soil water content [mmH$_2$O] at the end of the time period |
| | SURQ_CNT | Surface contribution [mmH$_2$O] to streamflow in the main channel during time step |
| | LATQ_GEN | Lateral flow generated in the HRU during time step [mmH$_2$O] |
| | GW_Q | Groundwater contribution to streamflow [mmH$_2$O], also called base flow |
| Sediment retention | SYLD | Sediment yield transported into the main channel during time step [t/ha] |

**Table 3: Model comparison.**

| Point of comparison | SWAT | RIOS | InVEST |
| --- | --- | --- | --- |
| **Model description** | Hydrologic model with different output variables | Implementation of activities to maintain, protect or restore ES | Different models for final and supporting ES used: Water Yield and SDR Model |
| **Inclusion of beneficiaries** | Cannot be included directly Visual comparison possible | Beneficiaries-raster to weight activity areas | Water yield model uses water demand table |
| **Uncertainty** | Calibration and valuation possible, but for study area no calibration | No option for calibration; comparison of input data with literature values | Calibration possible with sediment load or stream flow, but for study area no calibration |
| **Data requirements & preprocessing** | High data requirement and preprocessing | Medium data requirement and high preprocessing | Medium data requirement and low preprocessing |
| **Training effort** | Training effort high | Training effort medium to high | Training effort medium |
| **Time requirement** | High | Medium | Low to medium |

5  **Appendices**





**Appendix 1: SWAT soil input variables.**

| Variable | Definition | Obtained from or calculated by: |
|---|---|---|
| HYDGRP | Soil hydrological group (A, B, C, D), bases on infiltration characteristics | Determination suggested by Arnold et al. (2012), but required data not available; therefore simplified determination proposed by Environment and Natural Resources Trust Fund (n.d.) using soil texture provided by HWSD; |
| SOL_ZMX | Maximum rooting depth of the soil profile [mm] | Reference soil depth (REF_DEPTH from HWSD, 1000 mm), because no other data available |
| SOL_Z1,2 | Depth from soil surface to bottom of layer [mm] | Topsoil (300 mm) and subsoil depth (700 mm) of HWSD |
| SOL_BD1,2 | Moist bulk density [Mg/m³ or g/cm³] | Reference bulk density from HWSD calculated with equation from Saxton and Rawls (2006) (T/S_REF_BULK_DENSITY) |
| SOL_AWC1,2 | Available water capacity of the soil layer [mm $H_2O$/mm soil] | Calculated by SPAW – Soil Water Characteristics (Saxton and Rawls, 2009); |
| SOL_K1,2 | Saturated hydraulic conductivity [mm/hr] | Calculated by SPAW – Soil Water Characteristics (Saxton and Rawls, 2009); |
| SOL_CBN1,2 | Organic carbon content [% soil weight] | T_OC and S_OC from HWSD |
| SOL_CLAY1,2 | Clay content [% soil weight] | T_CLAY and S_CLAY from HWSD |
| SOL_SILT1,2 | Silt content [% soil weight] | T_SILT and S_SILT from HWSD |
| SOL_SAND1,2 | Sand content [% soil weight] | T_SAND and S_SAND from HWSD |
| SOL_ROCK1,2 | Rock fragment content [% total weight] | T_GRAVEL and S_GRAVEL from HWSD |
| SOL_ALB1 | Moist soil albedo | Calculated by the equation from Post et al. (2000): $Soil\ Albedo = 0.069 \cdot (color\ value) - 0.114$ with data from (Dijkshoorn et al., 2005); |
| USLE_K1 | USLE equation soil erodibility $K$ factor | Calculated with the equations of Williams (1995) given in Arnold et al. (2012) using the sand, silt, clay, and organic carbon content; |



**Appendix 2: RIOS and InVEST input raster data.**

| Data | Definition | Obtained from or calculated by: |
|---|---|---|
| **Average annual rainfall** | Mean annual rainfall depth in mm for each cell | WorldClim (Bioclimatic variables for tile 23), annual average from 1960-1990 (Hijmans et al., 2005) |
| **Rainfall depth or precipitation of wettest month** | Rainfall depth influences the amount of runoff of each cell. If the rainfall depth is not available, the mean precipitation of the wettest month can be used [mm] (Vogl et al., 2016) | WorldClim (Bioclimatic variables for tile 23), annual average from 1960-1990 (Hijmans et al., 2005) |
| **Mean annual actual evapotranspiration (AET)** | Mean annual values for each cell in mm | CGIAR-CSI, annual average from 1950-2000 (Trabucco et al., 2010) |
| **Reference evapotranspiration (only InVEST)** | Average annual reference evapotranspiration in mm being the potential loss of water from soil evaporation and alfalfa/grass transpiration if sufficient water is available; the equations (Penman-Monteith, Hargreaves etc.) for potential evapotranspiration (PET) are suggested, therefore the PET raster of CGIAR-CSI can be used | CGIAR-CSI, annual average potential evapotranspiration from 1950-2000 (Zomer et al., 2008) |
| **Rainfall erosivity** | Rainfall erosivity index $R$ depending on the intensity and duration of rainfall in $\frac{MJ \cdot mm}{ha \cdot hr \cdot yr}$ | Calculated using WorldClim data by an equation proposed by Mikhailova et al. (1997); |
| **Soil erodibility** | Soil erodibility $K$ measures the susceptibility of soil particles to erosion in $\frac{ton \cdot ha \cdot hr}{MJ \cdot mm \cdot ha}$ (Vogl et al., 2016) | Calculated using HWSD according to Sharp et al. (2016); |
| **Soil depth** | Average soil depth for each cell in mm | Obtained from HWSD data; |
| **Soil texture (only RIOS)** | Index value for each cell representing the soil texture class | Derived from HWSD; |
| **Plant available water fraction (only InVEST)** | *PAWC* is the fraction of water stored in the profile and available for plants' use (Vogl et al., 2016) | Calculated with SWAT input variable SOL_AWC (from |



| | | SPAW) by adding up the SOL_AWC values across horizons |
|---|---|---|
| **Location and number of beneficiaries (only RIOS)** | Location and number of beneficiaries depend on the chosen objective; different raster files for different objectives: erosion control: population density (rural and all beneficiaries); base flow and flood mitigation: population downstream (rural and all) | Calculated with population density (Alcaldía Municipal de León, 2008); |