# Peer review of "Modelling Hydrological Ecosystem Services – A state of the art model comparison"

_Hydrology and Earth System Sciences, 2017_

## Referee Comment (RC1) · S. J. Schymanski (Referee) · 23 Aug 2017

The manuscript is relatively well written, with nice graphical illustration of model outputs, but I was not able to extract any scientific insights from the manuscript. The manuscript reads a bit like a user manual for different models, not a scientific analysis. It does not contain any robust reality checks of the model outputs or insightful discussion of the scientific principles underlying the different models. Therefore, I would suggest to publish it as a technical report, but not a scientific paper. For a scientific paper, I was expecting so see clear science questions asked and answered, supported by evidence. For example on P5L17, you state: "RIOS is based on a science-based approach operating independently of scale or location and, therefore, it can be used at continental, country, or regional scale." This is a very interesting statement and I would

like to find out about the science in this approach, what it is based on, how it can operate independently of scale, what insights it produces and what are the uncertainties involved. Same for the other models.

---

## Author Comment (AC1) · 1 Sep 2017

The modelling of ecosystem services is a highly discussed topic among the scientific community. Different simulation models that have been developed in the past are still under a dynamic scientific debate. Several scientific publications (among them publications in the HESS journal; e.g. Doku et al. 2015) discuss or compare the applicability of models developed to quantify and map ecosystem services. The number of scientific publications dedicated to this topic illustrated the scientific significance of the assessment and comparison of ecosystem service model applications well. However, in the field of hydrology and water management traditional hydrological models have been applied to model (hydrological) ecosystem service. As stated in this publication, there are also numerous scientific papers on the application of traditional hydrological

models for ecosystem service modelling. Nevertheless, there are no scientific comparisons (documented, understandable and reproducible) of applications of specialize ecosystem service models AND traditional hydrological models for the modelling of hydrological ecosystem services for the same study area. This is what is essentially new in this publication. Moreover, the publication goes beyond simple model application but modifies the models in their pre- and post-processing to achieve comparable results among the different models and to increase their applicability in scientific problem contexts. The inclusion of ecosystem service beneficiaries in the application of the hydrological model is especially innovative. The modelling results of the three models and their comparison represent new scientific insights and may guide future modelling of hydrological ecosystems services. The focus of the model comparison is placed on applying the model for decision support, not on model theory. This means usability and applicability in decision-making context (e.g. for the establishment of payment schemes for ecosystem services) is regarded in this publication while discussions of the model's theory have been published elsewhere. The authors endorse the publication of this manuscript as a research article rather than a technical note.

---

## Referee Comment (RC2) · Anonymous Referee #2 · 3 Oct 2017

This is a review of the article entitled "Modelling Hydrological Ecosystem Services - A state of the art model comparison", recently submitted by Anna Lücke and Jochen Hack to Hydrology and Earth System Sciences (HESS). This article is a comparison of three pieces of software which provide assessment of hydrological ecosystem services. Two of them are based on hydrological modelling (with different levels of detail description). The first one uses the widely known SWAT model. The authors first compare the 'conditions of use' of the 3 pieces of software. Then a qualitative comparison of the results on a single case study (a rather small catchment in Nicaragua) is shown.

As hydrologist, I had limited my review to the hydrological issues.

In my opinion, the main audience (or at least, one of the main audiences) of HESS is among the community of hydrologists and hydrological modellers who do not neces-

sarily know much about this specific use (assessment of ecosystem services). If this is correct, then I have to warn the Editor that this article requires skills in the ecological sciences that many readers may not have.

Furthermore, I have not been able to understand what the scientific issue of this paper is. It provides a partial (incomplete) comparison of 3 pieces of software but there is no discussion about their relative scientific relevances. The article shows that some are more suitable for practical reasons but it does not assess if they provide sound results.

GENERAL COMMENTS ON THE METHODOLOGY

Much information is provided about the data used to set up or "feed" the "models". This is indeed valuable to ensure the reproducibility of the study. However on the contrary, there is no sound scientific descriptions of the 3 pieces of software: the theoretical grounds of the "models" are not given (for instance, the SWAT model is based on "widely accepted hydrological processes knowledge"; while this sentence is not sufficient to describe SWAT, it is quite a definitive statement, while some if not most readers of HESS might consider that some progresses are still to be made in our understanding of hydrological processes!). Therefore the methodology section is more like an end-user documentation ("cooking recipe") than a methodology description indicating the main assumptions and their consequences (potential flaws, bias...). As a consequence, it is difficult to discuss and understand the difference of the results of the 3 pieces of software (some examples are given in the detailed comments below).

One main methodological issue might concern the spatial and temporal resolution. The spatial resolution of the input data (rasters) is quite high (pixel size of 90 m x 90 m). The case study catchment is rather small (a few dozens of km2). The SWAT model computes on many HRUs. Its temporal resolution is not given in the article; I assumed it is chosen accordingly. However the 2 other pieces of software seem to work a yearly time step (or even longer time steps for RIOS), while providing high spatial resolution results. Since the spatial and temporal scales of hydrological processes which are

'behind' the ecosystem services ("simplifications of common hydrological relationships" for INVEST) are closely tied, are these descriptions and assessments relevant?

Furthermore, the chosen (hydrological environmental services) HES may be subject to significant seasonal variability, since some related hydrological processes are. The assessment time step (1 year) is therefore not relevant (this is stated in the results section, page 11). This is a quite big issue which may weaken the significance of the results. It would be useful to assess the potential impact of seasonality (in a quantitative manner).

GENERAL COMMENTS ON THE RESULTS

The article provides very few quantitative results: the models are only compared through maps on a single case study, which is far too little to allow a significant comparison. There is no hydrograph either (while all models, including SWAT, seems to be not calibrated). The reasons of similarities or differences are not really investigated. Furthermore, the results are compared one to each other but the methodology chosen by the authors doesn't allow them to provide an analysis of the relevance of the outputs of the 3 pieces of software (not even in a qualitative way).

Moreover, the HES are assessed at a pixel resolution or at the HRUs scale. However, even in such a small catchment, it is not clear to me whether the HES are (only) located at some specific places or may (also) be considered in a 'lumped' way. For instance, would 'upstream' HES benefit to downstream population? Then I am not sure how to interpret the results discussion on page 10 (lines 5-17).

CONCLUSIONS

Therefore I suggest the Editor to reject the article and invite the authors: (a) to submit their work to another publication more relevant for their research field, (b) or to make deeper the hydrological issues of their studies, (c) or to submit a much shorter "technical note" introducing the use of hydrological modelling for ecosystem services

assessment, which would refer to the relevant literature in some ecological sciences journals (including their own publication(s) id they also pick option (a)).

DETAILED COMMENTS

Page 1, line 21 ("Hydrological Ecosystem Services are defined here as proposed by Brauman et al. (2007)"): since many readers of Hydrology and Earth System Sciences may not be necessarily aware of the different definitions of HES and since they are likely not to read the journal within Brauman et al. (2007) published (journal dedicated to Ecology sciences), please give the definition of HES.

Page 2, line 4 ("free source"): do the authors mean "open source"?

Page 2, line 9 ("applicatin potential"): check spelling

Page 2, line 21 ("all modeling data [...]"): this is unclear; do the authors mean the input data?

Page 2, line 27 ("equal named municipality"): check English ("equalLY"?)

Page 3, line 15: a definition of "soil degradation" would be welcome.

Page 3, line 23 ("However the department of Leon [...]"): the logics is unclear to me; do the authors mean "moreover"?

Page 3, line 30 and following: I did not understand why so many details were necessary and important to the reader (even in order to reproduce the experiment). For example, what is the issue (sensitivity?) of the choice of the coordinate system (projection)?

Page 4, line 10 ("Sealed, major roads [...]"): why these data are important? Relevant? For a non specialist, it may look like as if all the available data were to be added in a large pot and "cooked" together.

Page 4, line 17: "the" traditional hydrological model doesn't exist so far (unfortunately ?). Our current knowledge and modelling skills do not let us to have a single model...

SWAT is one of the most used model for a specific application of hydrological modelling.

Page 4, line 29 ("The main components of SWAT"): this is too vague. Even if it is likely that most readers of HESS have met the SWAT model previously, some equations or some precise description of the hydrological processes are needed. Its time step is not given, while he spatial resolution is better discussed.

Page 5, line 5: again, the way the different terms taken into account in the water balance equation differs significantly from one temporal resolution to another...

Page 5, line 17 ("RIOS is based on a science-based approach"): what does it mean? Are some of the pieces of software studied here not science-based?

Page 5, line 18: again, I don't understand how the time steps (annual or longer) can be consistent with the spatial resolution and the hydrological processes (represented in RIOS in a simplified way).

Page 5, line 31 ("The transitions influence directly or indirectly hydrological processes [...]"): in my opinion, most HESS readers would be interested in many more details.

Page 6, line 2: Vogl et al. (2016) are quoted several times but I didn't manage to find it in the references section.

Page 6, line 19 ("the spatial explicit models of INVEST"): this sentence remains unclear to me.

Page 6, line 22 and following: the time step is a year while "the model calculates on a pixel basis"...

Page 7, line 20 (Sharp et al. 2016): this reference is not provided with the necessary information to find it easily (journal? Editor? URL?)

Page 7, line 20 and following: why raster and shape files definition is useful to the reader?

[Figure]

Page 7, line 33: a much finer time step is required to model sediment phenomena. Is there some kind of statistical approach ? Many more details would be welcome.

Page 7, line 34: a reference for RUSLE may be useful to some readers.

Page 8: subsection 2.3 may usefully be split in two parts (two subsections): one dedicated to the comparison of the input data and the approaches of the 3 "models" while the second one would only deal with the "post-processing" of the SWAT results in order to assess HES. Indeed, the latter is likely to particularly interest many HESS readers (how to use the outputs of a hydrological model, such as the SWAT model, in order to assess ecosystem services).

Page 9, lines 15 and 16: since Schmalz et al. (2016) published their results in an Ecology field journal, the authors may usefully provide (much) more details. In addition, why did the authors choose this particular approach?

Page 10, line 7: "visual comparison" is difficult and subjective, especially for readers who are not fully skilled in ecosystem services assessment. Some more quantitative results are needed.

Page 10, line 23 ("BECAUSE there are 2 different raster images representing all and rural beneficiaries, 2 simulations of RIOS are run"): this is unclear to me; isn't the main reason that the authors are interested in understanding the consequences for 2 different kind of beneficiaries?

Page 10, lines 25-28: are these pieces of information some "results" of the model simulation or are they direct consequences of the management rules implemented in RIOS?

Page 11, line 16 ("1,851,418 m3"): how many significant figures? Moreover, giving the results per subwatershed suggests that they are comparable (e.g. same superficies or same rainfall amount, etc.): is it so?

Page 11, line 24: same issue for the sediment retention.

Page 11, line 24 ("what is similar to the results of SWAT"): this may be usefully discussed.

Page 12, line 6 ("while SWAT and InVEST are based on similar conceptual approaches to model hydrological processes"): this has not been really shown (and I wonder if all hydrologists would agree in details).

Page 12, line 15 ("The chosen class limits have a strong influence on the results"): this seems to be a quite important issue: some quantitative results are needed to understand how much this is significant.

Figure 6: the legend is not sufficient.

Table 1: There is no streamflow data? (Therefore no calibration of the models). Furthermore, the temporal resolution of data (e.g. rainfall measures) would be usefully added.

---

## Referee Comment (RC3) · Anonymous Referee #3 · 5 Oct 2017

The authors present three models to assess Hydrological Ecosystem Services. Although the manuscript is well written and organised, I don't see the added value for the hydrological community. In my view, the authors just present the results of 3 models (which are not explained how they work) for a case study and then conclude that the results have different meaning for HES. Next, they come up with some guidelines to help to deal with those difference. This is too minor for a scientific article in HESS in my opinion. So maybe another journal/platform is better.

P1-L21: Please repeat the definition of HES of Brauman et al, since the hydrological community might not be well familiar with this concept.
P1-L21: Maybe explain the difference between HES and ES.

[Figure]

P2-L9: typo in 'application'

P4-L12-13: remove the line breaks before 'appendix'

P9-L14: typo 'form' => from

Fig 5: how does the postprocessing of SWAT works?

Fig 6: how should I interpret these results? What is meant by 'capacity of ES' and 'beneficiaries'?